# Paving the Way towards an Armenian Data Cube

**Shushanik Asmaryan [1],\*** , **Vahagn Muradyan [1]**, **Garegin Tepanosyan [1]**, **Azatuhi Hovsepyan [1]**,
**Armen Saghatelyan [1]**, **Hrachya Astsatryan [2]**, **Hayk Grigoryan [2]**, **Rita Abrahamyan [2]**,
**Yaniss Guigoz [3,4]** and **Gregory Giuliani [3,4]**

[1] GIS and Remote Sensing Department, Center for Ecological-Noosphere Studies NAS RA,
   Yerevan 0025, Armenia
[2] Institute of Informatics and Automation Problems NAS RA, Yerevan 0014, Armenia
[3] Institute for Environmental Sciences, University of Geneva, 1205 Geneva, Switzerland
[4] United Nations Environment Programme/Global Resource Information Database (UNEP/GRID-Geneva),
   1219 Geneva, Châtelaine, Switzerland
[\*] Correspondence: shushanik.asmaryan@cens.am; Tel.: +374-9400-8214

**Abstract:** Environmental issues become an increasing global concern because of the continuous pressure on natural resources. Earth observations (EO), which include both satellite/UAV and in-situ data, can provide robust monitoring for various environmental concerns. The realization of the full information potential of EO data requires innovative tools to minimize the time and scientific knowledge needed to access, prepare and analyze a large volume of data. EO Data Cube (DC) is a new paradigm aiming to realize it. The article presents the Swiss-Armenian joint initiative on the deployment of an Armenian DC, which is anchored on the best practices of the Swiss model. The Armenian DC is a complete and up-to-date archive of EO data (e.g., Landsat 5, 7, 8, Sentinel-2) by benefiting from Switzerland's expertise in implementing the Swiss DC. The use-case of confirm delineation of Lake Sevan using McFeeters band ratio algorithm is discussed. The validation shows that the results are sufficiently reliable. The transfer of the necessary knowledge from Switzerland to Armenia for developing and implementing the first version of an Armenian DC should be considered as a first step of a permanent collaboration for paving the way towards continuous remote environmental monitoring in Armenia.

**Keywords:** big earth data; sustainable development goals; swiss DC; Armenian DC; Landsat; sentinel; analysis ready data

---

## 1. Introduction

Environmental problems become an increasing global concern continuously put stress on natural resources. Global challenges with environmental compartments dimensions such as fresh water, air quality, deforestation, land management or urbanization require improved and updated information, which acquired the dynamic nature of environmental conditions [1,2]. Earth observations (EO) data (satellite and in-situ), provide strong monitoring mechanisms for above mentioned environmental problems because of their geospatial consistency, accessibility, repeatability, and global coverage [3,4]. It proves that by providing a summarized view of a given spatial extent remotely sensed EO becomes an important element to monitor the ecological state of the different environmental compartments (water, soil, plants, etc.). So, precise and reliable data are an important component of the environmental monitoring systems [5]. There are several open remote sensing (RS) data repositories that provide highly valuable, timely and precise remotely sensed EO information. However, there is a strong need of a set of geoprocessing tools, which would allow to retrieve the full information potential of

EO data [6–8]. This is mainly because of EO data complexity, large-volume, and deficiency of good processing capacities [8–10].

Considering EO data as part of Big Data, because of their volume (e.g., Landsat archive is 7.5PB), variety (e.g., optical, radar), and velocity (e.g., Sentinel data temporal resolution is every 5 days), innovative tools are required to reduce the time and knowledge needed to access, prepare and analyze large volumes of EO data having steady and spatially adjusted calibrated observations [5].

EO Data Cube (DC) is a new paradigm aiming to meet Big Earth Data challenge as a new approach to store, organize, manage and analyze EO data [11,12].

Hence, Data-Cube is now considered as a promising technology to perform time-series analyses of large satellite Analysis Ready data-sets like Landsat and Sentinel [13].

There are several operational DC initiatives, covering different spatial scales and storing different data, using different infrastructures and software implementations (e.g., Earth Observation DC (EODC—http://eodatacube.eu), Earth on Amazon Web Services (EAWS—https://aws.amazon.com/earth/), Google Earth Engine (GEE—https://earthengine.google.com), Earth System DC (ESDC—http://earthsystemdatacube.net) [5].

As of end 2018, three countries (Australia, Switzerland, and Colombia) have DC on a national-scale (https://www.opendatacube.org/ceos).

Australian Geoscience DC (AGDC—http://www.datacube.org.au), renamed as Digital Earth Australia, was the first successful attempt, making entire continent's geographical datasets available to researchers and policy-makers [12,14]. Lessons learned from design and implementation of AGDC underpin Chinese DC (CDC) based on the new Open Geospatial Consortium (OGC) Discrete Global Grids System (DGGS) standard and cloud computing technologies and Colombian DC [15,16].

However, Switzerland is the second country in the world, which claimed to have a national-scale EODC. The Federal Office for the Environment supports the Swiss DC (http://www.swissdatacube.ch). It is developed, implemented and operated by the UN Environment (UNEP)/GRID-Geneva in partnership with the University of Geneva [5]. Currently, the Swiss DC contains 35 years of Landsat 5,7,8 (1984–2019), four years of Sentinel-2 (2015–2019), and 5 years of Sentinel-1 (2014–2019) Analysis Ready Data over Switzerland (total volume: 6TB; 200 billion observations) [17].

The Committee of Earth Observation Systems (CEOS) has vision, that more over 20 countries will be developing and realizing their Data-Cube infrastructure by 2022 [18].

Armenia is among these countries, aiming to gain the knowledge and to exchange experience from Switzerland implementing its own DC for several reasons: (i) Armenia still faces numerous environmental challenges as one of the most industrialized post-soviet countries; (ii) since the 90s, the economic policy moved towards supporting industrial development mainly ignoring environmental interests; (iii) in 2016, Armenia had initiated the Sustainable Development Goals (SDG) nationalization process and still face-off various problems caused by the lack of sufficient data hindering efficient national environmental monitoring; (iv) alternative ways need to be developed and realized to fill this gap and EODC represents a promising solution.

The paper aims to present the Swiss—Armenian joint initiative on the deployment of an Armenian DC, which is anchored on the best practices of the Swiss model.

## 2. Building the Armenian DC

Armenia was among the selected countries to contribute towards the shaping of the global development agenda, which was both a privilege and recognition of the country's unique perspective on development [19,20]. However, when monitoring the process of attaining several SDG targets (e.g., SDG target 6.6; SDG target 15.3; SDG target 15.4) an important problem of data disaggregation was encountered. EO can support the data aggregation process by providing policy makers with repeatable, continuous and multi-annual series of quantitative and qualitative data. The integration of EO technologies into decision making process is still to be improved in Armenia. So far, Armenian "decision makers" rely on the data provided by a few research or international organizations, which are

experienced in working with EO data and technologies [21–23]. Taking into account the fact that reliable remotely sensed monitoring of the identified SDGs requires EO systems allowing systematic acquisitions, free and open-access data and high quality imagery, the Landsat and ESA's Sentinel missions are the main data sources used. But high performances computational resources are needed to maintain process, visualize and share the EO-based monitoring data. It could be done by creating linkages with new platforms such as DCs that empower data visualization by providing an easier way to visualize environmental changes.

Thus, Swiss-Armenian cooperation initiated the establishment of the Armenian DC as a full and updated archive of EO data (e.g., Landsat, Sentinel), benefiting from the experience of the University of Geneva in implementing the Swiss DC.

In order, to transfer the necessary knowledge, it is vital to develop new capacities. This helps to reach adoption, acceptance and commitment to this new technology for increasing the capacity to access and use Earth Observations [24]. Capacity development can be defined as "human, scientific, technological, organizational, and institutional resources and capabilities" to "enhance the abilities of stakeholders to evaluate and address crucial questions related to policy choices and different options for development" (GEO Secretariat 2006). Three levels of capacity building can be defined: (1) human (e.g., education and training); (2) institutional (e.g., improving the comprehension of the value of geospatial data for decision-making); and (3) infrastructure (e.g., installing/configuring/managing of the technology). This should help demonstrating the benefits of EODC through appropriate examples and best practices to strengthen: (1) existing observation systems; (2) capacities of decision-makers to use it; and (3) capacities of the general public to understand important environmental, social and economic issues at stake. Such initiatives can also be beneficial for providers to increase their visibility and reliability nationally and internationally by participating in the approach to build such systems [25,26].

Recognizing these needs and based on the experience acquired in developing capacity building material for implementing Spatial Data Infrastructure [24] similar to the Bringing GEOSS Services into Practice, the Swiss team started to develop an integrated set of teaching material and software to give the necessary knowledge to efficiently install, manage and use an EODC based on the Open DC software stack.

The successfully installed Armenian DC is already available via http://datacube.sci.am (Figure 1) and the "Bringing Open Data Cube into practice" material is available at: http://www.swssidatacube.ch/products.

As in the case of the Swiss DC, a fundamental aspect when building a DC is to have Analysis Ready Data products, ingested, stored and available in the database. Analysis Ready Data (ARD) are concerned by the four first steps (data acquisition, radiometric calibration, conversion to top of atmosphere(TOA) reflectance and Surface reflectance) allowing then to analyze data and generate time-series [5]. All procedures of discovering, downloading from different repositories (e.g., ESPA, Sentinel Data Hub) and preprocessing were planned to be automated as much as possible and should be interoperable.

Thus, the Armenian DC contains 3 years (2016–2019) of Landsat 7 and Sentinel-2 analysis ready data over Armenia.

The full coverage of Armenia includes 11 Sentinel-2 (38TLL, 38TML, 38TNL, 38TLK, 38TMK, 38TNK, 38SMJ, 38SNJ, 38SPJ, 38SNH, 38SPH) and 9 Landsat 7 (171031, 170031,169031, 171032, 170032, 169032, 168032, 169033, 168033) scenes. It requires around 30–40 min to download and process a single Sentinel-2 image. The system deployment environment is Ubuntu server version 18.04 with 64-bit virtual machine, 64GB of RAM, 8 cores and a storage space of 2 TB. For downloading the correct scenes of our region, the boundary and projection conditions are provided, after which the datacube platform allows to download the available satellite images from global databases and translate data from the Earth observation satellites into ready-to-use insights about the continent's environmental conditions. Armenia is located inside a rectangle with the upper left (38.32335165219022, 42.98858178626198) and lower right (41.551890393271684, 47.320774961261485) points in the Earth coordinate system.

The Armenian DC uses the National e-infrastructure, which is a complex IT infrastructure consisting of both communication and distributed computing infrastructures [16].

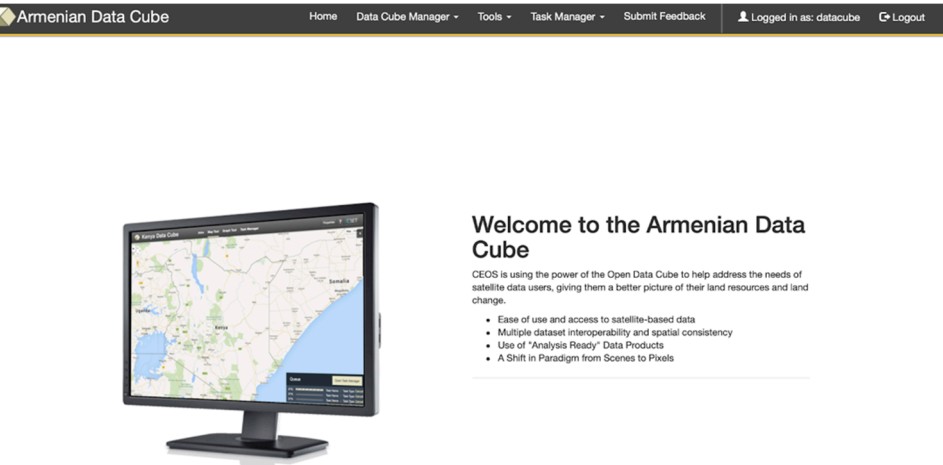

**Figure 1.** Interface of the Armenian DC.

## 3. Discussion

### 3.1. Lake Sevan as a Case Study: A Problem Statement

Among the issues where Armenian DC could provide a set of excellent tools and being demonstrated as a disruptive technology is the monitoring of the shoreline changes of Lake Sevan.

Lake Sevan is one of the most ecologically sensitive areas in Armenia. Since the beginning of the last century, the shoreline of this biggest freshwater lake in Armenia and South Caucasus has been changing continuously with different intensity causing many ecological problems: eutrophication of the lake, activation of erosion processes and so forth [27]. This makes it urgent to study the shoreline changes in order to understand the effects these produce on the near-shore belt [28,29].

Mapping and detection of coastline changes from satellite images have become increasingly important over recent decades, especially because satellites capture and provide data in visible and infrared spectral bands where the land and water can be easily distinguished [30–32]. These make optical satellite images containing visible and infrared bands of the electromagnetic spectrum widely used for coastline mapping especially when these images are easily obtainable [33].

There are several studies where the satellite optical imagery was used to assess Lake Sevan water quality [34–36]. However, there is no direct study on detecting changes of the Lake coastline using time-series analysis of satellite EO data and it is easy to perform if the data is openly available. The satellite image analysis enables to study the water boundary changes using the water detection service provided in the Armenian DC platform (Figure 2).

Exploration of the full potential of EO data requires huge computing resources enriched by specialized algorithms and tools [5]. So following Australian and Swiss experience on DC Swiss-Armenian research group decided to develop an automatized tool for shoreline delineation in ADC.

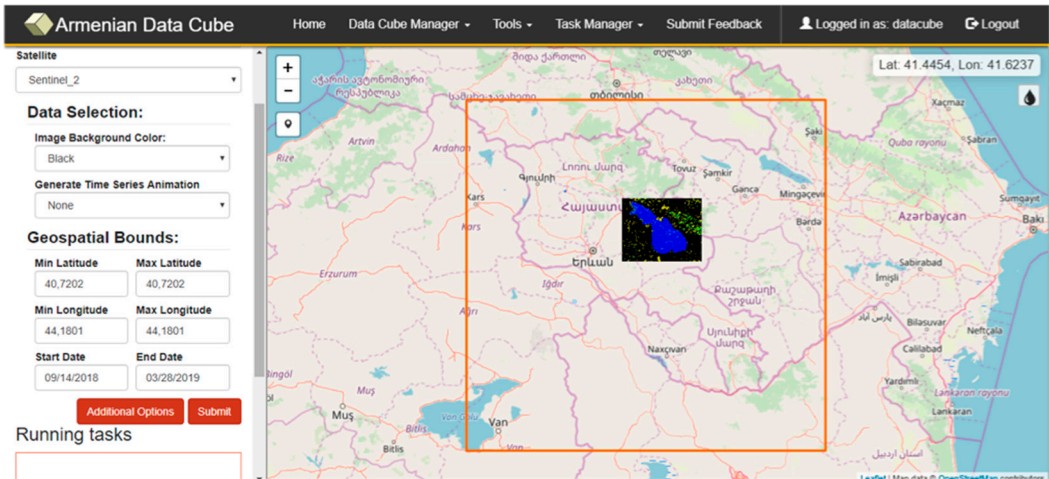

**Figure 2.** Sentinel-2 ingestion of Lake Sevan.

*3.2. Analysis Ready Data Production (Data Availability, Access, Ingestion Preprocessing)*

The main phase when developing DC is the preparation of ARD allowing to analyze data and generate time-series [5].

Satellite data scenes (Landsat 5,7,8 and Sentinel-2) were accessible via gsutil: a Python application, which gives an access to Google Cloud Storage from Command lines (https://cloud.google.com/storage/docs/gsutil).

The Live Monitoring of Earth Surface (LiMES) framework has been used for ARD preparation, which is a framework that helps to automate EO data discovery and (pre-) processing using interoperable set of tools transforming observations into the information products applicable for monitoring environmental changes. This framework is developed using a system of large storage capacities, high performance distributed computers, and interoperable standards to develop a scalable, coherent, flexible, and efficient analysis system, which can be used on various domains through decades of data for monitoring [5].

*3.3. Image Processing*

There are several methods of water object identification and shoreline delineation, which include classification and spectral signature feature analysis, which divided into single-band and multi-band methods [37].

Single band and multi-band threshold methods are widely used in optical RS to extract water bodies [38].

Single band method is a simple approach allowing to extract water surface information. Multi-band threshold methods are based on comprehensive consideration of each band and are widely used in water body extraction.

McFeeters [39] Normalized Difference Water Index (NDWI), which is well-known band-ratio method, which has been studied and used in the experiments via Python scripting with Sentinel-2 and Landsat scenes (Figure 3).

$$NDWI = (G - NIR)/(G + NIR) \tag{1}$$

It uses green (G) and near-infrared (NIR) spectral bands to maximize water feature identification (1). McFeeters proposed a zero threshold to separate water other land.

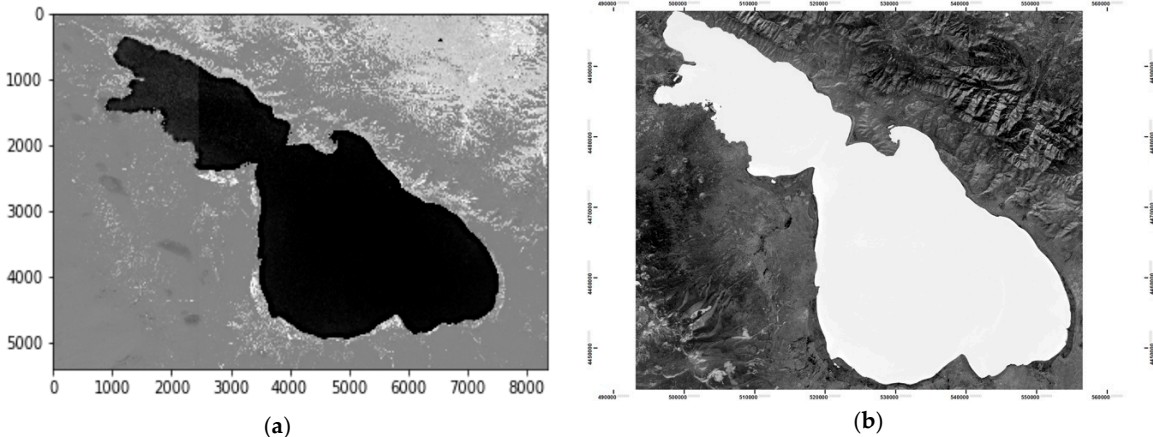

**Figure 3.** McFeeters band-ratio algorithm for Lake Sevan calculated from (**a**) Sentinel-2 and (**b**) Landsat sensors.

*3.4. Validation of the Results*

The validation step is an important component and not a straightforward task as DC has number of limitations, among which unknown quality of the automatized geoprocessing results using different verification algorithms. It comprises multiple components ranging from in situ measurements collection, modeling and retrieval of land surface variables to scale related analysis [40]. All these factors complicate the validation issue.

The validation step is an essential component and not a straightforward task, as the DC platform may generate unknown quality of the automatized geoprocessing results using different verification algorithms. It comprises multiple components ranging from in situ measurements collection, modeling and retrieval of land surface variables to scale related analysis [40]. All these factors complicate the validation issue.

The experimental verification of McFeeters band math (NDWI) calculation results was performed integrating the results of diverse observation, such as high-resolution remote-sensing products (UAV imagery) received during field campaign held in 2018 using Sensefly eBee and the hydrological data provided by the Service of Hydrometeorology and Active Influence on Atmospheric Phenomena SNCO, Ministry of Emergency Situation of Armenia (hereafter Service).

The shorelines derived via NDWI from Sentinel-2 and Landsat 8 were compared with the shoreline received from UAV for the small portion of the north-east shore (2 km).

To a first approximation the visual comparison of shorelines derived via NDWI from Sentinel-2 and Landsat 8 and UAV image shows that they match quite well despite the differences of spatial resolutions Landsat 8 (30 m), Sentinel-2 (10 m), UAV image (30 cm) (Figure 4).

The other approach was to compare the surface areas derived using NDWI from Sentinel-2 (12 January 2015) and Landsat 8 (29 December.2015) with the surface areas measured and calculated by the Service on 1 January.2015 and 1 January.2016 respectively (Table 1). The Table 1 shows that the differences between provided surface areas are 6.38 sq.km and 9.16 sq.km for Sentinel -2 and Landsat 8 respectively. It should be stressed that the images selected for comparison were acquired near the time of the hydrological data measurements.

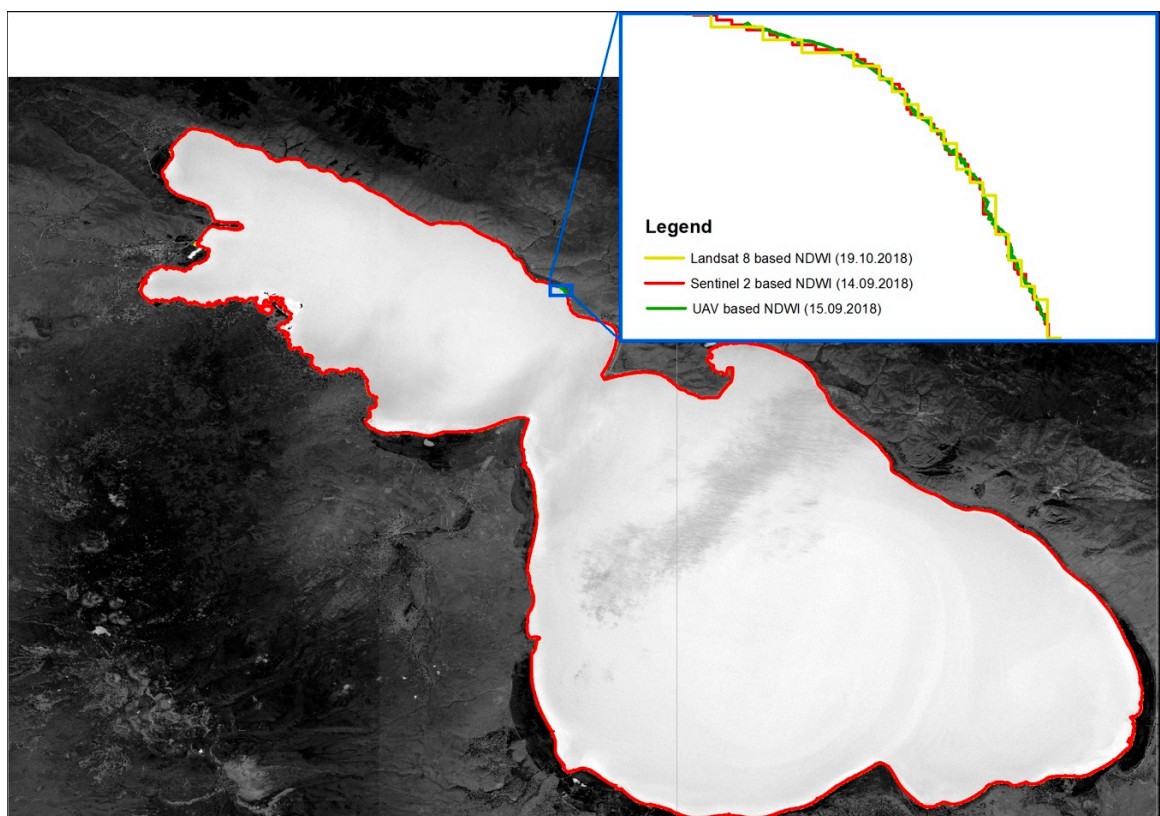

**Figure 4.** The comparison of shorelines derived from UAV, Sentinel-2 and Landsat imageries.

**Table 1.** The area of Lake Sevan according to Service and NDWI.

| Sensor | Total Area (sq.km) | | |
|---|---|---|---|
| | Satellite Image | Service | The Difference |
| Sentinel-2A (12 January 2015) | 1269.18 | 1275.56 | 6.38 |
| Landsat 8 OLI (29 December 2015) | 1265.83 | 1274.99 | 9.16 |

## 4. Conclusions

This paper aimed to present the international Swiss—Armenian joint initiative to deploy next national DC in Armenia, which becomes the fourth national DC in the world after Australia, Switzerland and Cambodia. ADC is one of the best applications of the Armenian national e-infrastructure, which should be updated and empowered continuously in order to reveal the full potential of this innovative technology.

Thus far, the ADC is enriched with complete and up-to-date archive of EO data and successfully works for the simplest issues such as delineation of Lake Sevan.

Landsat and Sentinel image-based delineation of shorelines using NDWI spectral index gives sufficiently reliable results for Lake Sevan.

It should be added that the web-based User Interface has been developed by CEOS [41] to allow users exploring the mains functionalities of the data cube. However, for developing more advanced/tailored applications or services, the Python Application Programming Interface (API) is the preferred choice.

Once the Armenian Data Cube will be fully operational and will generate "official" products, they will be complied with the FAIR (Findable, Accessible, Interoperable, Re-usable) data principles [42,43], which will include adding a license such as Creative Commons and having a Digital Object Indentifier (DOI) for each generated datasets/products.

It could be stressed that Armenian DC has a potential to transform the EO into useful information for users and represents a prospective solution for remotely sensed environmental monitoring in Armenia. So the analysis between Armenian and Swiss DC and the transfer the necessary knowledge from Switzerland to Armenia for developing and implementing the first version of an ADC should be continued paving a way towards continuous remote environmental monitoring in Armenia.

**Author Contributions:** Conceptualization, S.A., A.S., H.A., Y.G. and G.G.; Data curation, V.M., G.T., H.G. and R.A.; Funding acquisition, S.A., H.A. and G.G.; Investigation, S.A., V.M., G.T., A.H., H.G. and R.A.; Project administration, S.A., H.A. and G.G.; Supervision, G.G.; Validation, V.M., G.T. and A.H.; Visualization, Y.G.; Writing—original draft, S.A., H.A., H.G., Y.G. and G.G.; Writing—review & editing, S.A., V.M., G.T., A.H., A.S., H.A., R.A., Y.G. and G.G.

**Funding:** This research was funded by the UNIVERSITÉ DE GENÈVE as the Leading House (hereinafter referred to as "LH") for the bilateral Science and Technology cooperation program with Russia and the CIS Region, Grant number "SFG 163".

**Acknowledgments:** The research was supported by the RA MES State Committee of Science and Russian Foundation for Basic Research (RF) in the frames of the joint research project "SCS 18RF-140" and "RFBR 18-55-05015 Arm-a" accordingly. The authors would like to also thank the Swiss Federal Office for the Environment (FOEN) for their financial support to the Swiss Data Cube. Methodologies used in this publication partly rely on the Swiss Data Cube (http://www.swissdatacube.org) material and methodologies, operated and maintained by UN Environment/GRID-Geneva, the University of Geneva, the University of Zurich and the Swiss Federal Institute for Forest, Snow and Landscape Research WSL.

**Conflicts of Interest:** The authors declare no conflict of interests.

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
