# Peer review of "Paving the Way towards an Armenian Data Cube"

_data_

Round 1
Reviewer 1 Report
This is an overall very readable paper. Presentation is logical and clear. There is not very much scientific content, as it is an implementation of Data Cube. The consistency of presentation can be improved a little. Please see below.
Detailed comment:
Line 22 and elsewhere: There shouldn't be spaces around the "-" in "Swiss-Armenian" as in "Swiss-Armenian joint initiative.
Line 25 and elsewhere: The use of acronym is inconsistent. For example, both "Swiss DC" and "SDC" have been used repeatedly in the text. To improve readability, it is recommended to use "Swiss DC" and "Armenian DC" for "Swiss Data Cube" and "Armenian Data Cube" respectively. Further shortening serves no gain and diminishes readability, especially because SDC and SDG (Sustainable Development Goals) are too similar.
Line 94. Why is "Agenda" capitalized?
Figure 3 caption. In the text, it is referred to as "McFeeters band-ratio algorithm/method". Why change it to "band math"? They should be consistent.
Line 216: The spatial analytic platforms such as DC has a number of limitations, among which is the unknown quality of ..." Bold-faced texts should be inserted.
Line 229: It is recommended to replace "though" with "despite".
Author Response
Point1: This is an overall very readable paper. Presentation is logical and clear. There is not very much scientific content, as it is an implementation of Data Cube.
The consistency of presentation can be improved a little.
Response1: Thank you
Detailed comments:
Point2: Line 22 and elsewhere: There shouldn't be spaces around the "-" in "Swiss-Armenian" as in "Swiss-Armenian joint initiative.
Response2: Thank you. We changed it in the text.
Point3: Line 25 and elsewhere: The use of acronym is inconsistent. For example, both "Swiss DC" and "SDC" have been used repeatedly in the text. To improve readability, it is recommended to use "Swiss DC" and "Armenian DC" for "Swiss Data Cube" and "Armenian Data Cube" respectively. Further shortening serves no gain and diminishes readability, especially because SDC and SDG (Sustainable Development Goals) are too similar.
Response3: Thank you. We agree with the reviewer. We have made corrections.
Point4: Line 94. Why is "Agenda" capitalized?
Response4: Thank you. We have corrected.
Point5: Figure 3 caption. In the text, it is referred to as "McFeeters band-ratio algorithm/method". Why change it to "band math"? They should be consistent.
Response5:Thank you. We have corrected.
Point6: Line 216: “The spatial analytic platforms such as DC has a number of limitations, among which is the unknown quality of ..." Bold-faced texts should be inserted.
Response6: We thank the reviewer. We adopt the text as follow:
“The validation step is an essential component and not a straightforward task, as the DC platform may generate unknown quality of the automatized geoprocessing results using different verification algorithms. It comprises multiple components ranging from in situ measurements collection, modeling and retrieval of land surface variables to scale related analysis [40]. All these factors complicate the validation issue.”
Point7: Line 229: It is recommended to replace "though" with "despite".
Response7:Thank you. We have corrected.
Reviewer 2 Report
Lines 72 & 243: Columbia versus Cambodia, likely not Cambodia.
What are the plans to put a license on these data, e.g. Creative Commons? Seems it would be hard to cite these data.
I had a hard time getting much of anything to work on the project interface. Many different challenges around latitude, longitude, and resolution specifications.
Author Response
Point1: Lines 72 & 243: Columbia versus Cambodia, likely not Cambodia.
Response1: Thank you. We have corrected.
Point2: What are the plans to put a license on these data, e.g. Creative Commons? Seems it would be hard to cite these data.
Response2: We thank the reviewer for pointing this issue. Concerning the data used of the paper, they are used for illustration purposes. However, once the Armenian Data Cube will be fully operational and will generate “official” products, our plans are to comply with the FAIR (Findable, Accessible, Interoperable, Re-usable) data principles [REF1], [REF2]. This will include adding a license such as Creative Commons and having a Digital Object Indentifier (DOI) for each generated datasets/products.
REF1: M. D. Wilkinson et al., “The FAIR Guiding Principles for scientific data management and stewardship,” Scientific Data, vol. 3, p. 160018, Mar. 2016. DOI: 10.1038/sdata.2016.18
REF2: S. Stall et al., “Make scientific data FAIR,” Nature, vol. 570, no. 7759, p. 27, Jun. 2019. DOI: 10.1038/d41586-019-01720-7
Point3: I had a hard time getting much of anything to work on the project interface. Many different challenges around latitude, longitude, and resolution specifications.
Response3: Thank you for pointing this issue. Still, we would inform that, the web-based User Interface has been developed by CEOS [REF1] to allow users exploring the mains functionalities of the data cube. However, for developing more advanced/tailored applications or services, the Python Application Programming Interface (API) is the preferred choice.
REF1: The Ceos Data Cube Portal: a User-Friendly, Open Source Software Solution for the Distribution, Exploration, Analysis, and Visualization of Analysis Ready Data
https://ieeexplore.ieee.org/document/8518727
Reviewer 3 Report
I think the paper can be published in Data.
Author Response
Point1: I think the paper can be published in Data.
Response1: Thank you